# Cognitively Biased Users Interacting with Algorithmically Biased Results in Whole-Session Search on Debated Topics

## ABSTRACT

When interacting with information retrieval (IR) systems, users, affected by confirmation biases, tend to select search results that confirm their existing beliefs on socially significant contentious issues. To understand the judgments and attitude changes of users searching online, our study examined how *cognitively biased* users interact with *algorithmically biased* search engine result pages (SERPs). We designed three-query search sessions on debated topics under various bias conditions. We recruited 1,321 crowdsourcing participants and explored their attitude changes, search interactions, and the effects of confirmation bias. Three key findings emerged: 1) most attitude changes occur in the initial query of a search session; 2) Confirmation bias and result presentation on SERPs affect the number and depth of clicks in the current query and perceived familiarity with clicked results in subsequent queries; 3) The bias position also affects attitude changes of users with lower perceived openness to conflicting opinions. Our study goes beyond traditional simulation-based evaluation settings and simulated rational users, sheds light on the mixed effects of human biases and algorithmic biases in information retrieval tasks on debated topics, and can inform the design of *bias-aware* user models, human-centered bias mitigation techniques, and socially responsible intelligent IR systems.

## CCS CONCEPTS

• **Information systems** → **Users and interactive retrieval**.

## KEYWORDS

Confirmation bias, Interactive information retrieval, Credibility and usefulness evaluation, Bounded rationality, Search Session

**ACM Reference Format:**

Anonymous Author(s). 2024. Cognitively Biased Users Interacting with Algorithmically Biased Results in Whole-Session Search on Debated Topics. In *Proceedings of ACM Conference (Conference'17).* ACM, New York, NY, USA, 11 pages. https://doi.org/10.1145/nnnnnnn.nnnnnnn

## 1 INTRODUCTION

In contrast to the assumptions of most formal user models, information searchers are *boundedly rational* and their judgments and search behaviors are often affected by both *system biases* and their *cognitive biases* [37]. When encountering information involving multiple or contradicting perspectives, people tend to accept the

information that is consistent with their own beliefs [69], leading to higher risks of opinion polarization and misinformation spreading [2, 42]. This preference for confirming existing opinions is called *confirmation bias* [46]. This bias was also observed in interactive information retrieval (IIR) research, with users frequently clicking attitude-confirmation results while avoiding attitude-disconfirming results [69–71]. Such interactions may reinforce users' existing opinions due to the *exposure effect*, depending on the amount of time and frequency of the interactions on a specific opinion [3, 18]. Consequently, this effect may lead to biased judgments and unfair decisions, and further generate biased training data for search systems, which may aggravate societal biases through personalized retrieval and recommendations [24, 51, 52].

However, most studies only considered human biases in ad hoc retrieval and isolated document evaluation contexts [e.g., 18, 55, 59, 71]. The single-query session considers each search as a standalone query but ignores users' interactions and cognitive changes in prolonged task-driven search sessions [41]. However, the effects of confirmation bias in *multi-query* sessions on users' attitudes or search interactions still remain understudied. In addition, as mainstream research tends to examine human bias and algorithmic bias *separately* (with human biases being studied in user studies, while algorithmic bias is mainly being investigated in simulation-based experiments), it is not clear how cognitively biased users would react to and make judgments under biased SERPs.

Inspired by previous work on user attitudes and interactions in searching debated topics [18, 55], this study designed a simulated three-query search session under various *bias conditions*. The bias condition is represented by the result presentations on SERPs (i.e., manipulated ranks, amounts, or obfuscation of opinionated results on SERPs) and the combinations of SERP conditions in the three-query session. We conducted a crowdsourcing study and assigned participants to different bias conditions for between-subjects experiments. We investigated users' attitude changes, search interactions, and confirmation bias effects in multi-query search sessions.

Our study defined and investigated two types of *within-session* attitude changes, including the **accumulative attitude change**, defined as the absolute sum of the differences between *pre-* and *post-query* attitudes; and the **directional attitude change**, defined as the difference between preexisting and the most recent post-query attitudes, considering the preexisting attitude's direction. Initially, we explored the nature of within-session attitude changes, focusing on their relationships with user characteristics and query-wise attitude changes (i.e., differences between pre- and post-query attitudes). Subsequently, we examined how confirmation bias influences search interactions. We analyzed the effects of SERP conditions, including *within-query effects* (the immediate impact on the current query) and *between-query effects* (the extended impact of previous queries on a subsequent query) on search interactions. Additionally, we examined how bias levels and

the sequence position of biased SERPs in the session affect attitude changes. Our findings are threefold:

- Users' most attitude changes occur in the first query in a search session, while some users' attitudes could still be affected in the second and third queries.
- Confirmation bias and algorithmically biased SERP presentation affect the number and depth of clicks in the current query and users' perceived familiarity with clicked results in subsequent queries.
- Users with lower perceived openness (i.e., a self-reported value about reactions to conflicting opinions) are more susceptible to directional attitude changes when influenced by the sequence position of biased queries.

Overall, these findings enhanced our knowledge about biases on both human and system sides at the session level. Our study urges search system designers to consider user characteristics and result presentation to effectively intervene in users' search behaviors and prevent bias reinforcement. Our study further underscores the need for responsible and transparent application of implementing bias mitigation strategies, emphasizing the importance of prioritizing users' well-being and providing accurate and unbiased information.

## 2 RELATED WORK

To solidify the justification for our work, this section introduces previous research on cognitive biases [e.g. 31, 63], especially confirmation bias, as well as algorithmic bias and mitigation in IR.

### 2.1 Confirmation Bias in IR

During search processes, human cognitive biases can significantly affect search performances, making them less effective and accurate, and further affect the success of whole-session search tasks [3, 27, 28, 36, 44, 47, 57, 72]. IR researchers have investigated these biases to understand seemingly irrational decisions in users' search activities and perceptions of search results [e.g. 3, 19, 36]. Previous studies showed that certain cognitive biases, stemming from pre-existing beliefs or ongoing sessions, can influence user behaviors, perceptions of the relevance and usefulness of search results, and their whole-session satisfaction levels [38]. Some prominent biases include anchoring bias [10, 61], reference-dependent effects [9, 38], user expectations [40, 65, 66], and confirmation biases [69, 70].

*Confirmation bias* is one of the most common cognitive biases that users face, which refers to the tendency to search for, interpret, and remember information that confirms one's preexisting beliefs or attitudes [46]. This bias can occur in searches on debated or controversial topics and shape users' opinions and search behaviors [35, 69]. White and his colleagues [70, 71] have conducted several studies on belief dynamics and biases in web search, and their findings indicate that confirmation bias can persist even when users were presented with evidence that contradicts their beliefs. Pothirattanachaikul et al. [51] conducted a study to investigate the effects of document opinion and credibility on confirmation bias in IR. They found that users were more likely to click on search results that confirmed their preexisting beliefs and attitudes, and the document's opinion and credibility also affected their beliefs. Draws et al. [18] designed different ranking conditions and analyzed the confirmation bias and the search engine manipulation effect (SEME)

in searching debated topics. SEME refers to the situation where users tend to accept the opinions in the top documents because of the biased ranking [20]. They recruited participants with neutral or weak opinions, examined their actively open-minded thinking, and explored whether user cognitive engagement (e.g., willingness, inclusiveness, tolerance) would increase their attitude changes, as reported in previous research [48, 64, 70]. Although their results did not reveal differences in attitude changes among the ranking conditions or user groups, they suggested potential exposure effects on attitude changes. The exposure effect depends on the amount of time and frequency a user is exposed to a specific opinion, which may reinforce their attitudes toward that opinion [3].

### 2.2 Bias Reinforcement And Mitigation

The effect of confirmation bias in IR can have a complex process and significant consequences. Firstly, users might be exposed to biases in search results with algorithmic biases originating from the training data and retrieval algorithms (e.g., presenting results with ranking or diversity bias and information with unfairness and inequity) [22, 34, 62]. Then, users influenced by algorithmic biases or unfair results may form stereotypes and opinionated or polarized beliefs [8, 32, 68]. In addition, users' influenced search behaviors can further generate biased data for the search system, leading to more personalized and biased results. Ghenai et al. [26] conducted a think-aloud study and found that biased search engine results can significantly influence users' decisions based on the majority of the search results. Pogacar et al. [50] conducted an online survey and found that biased search results towards incorrect information can lead to incorrect decisions. Knobloch-Westerwick et al. [33] found that users preferred attitude-consistent messages and high-credibility sources. They also suggested the exposure effects of diverse search results on reinforcing or mitigating attitude strength.

Search result diversity plays a vital role in promoting fairness by encompassing a variety of topics and perspectives [23, 25]. This diversity contributes to topical fairness as it ensures that users are exposed to a broad range of information items and viewpoints [16, 67]. Fairness in IR can take different forms, such as demographic parity, which emphasizes the equitable presentation of search results for various user groups [62]. In addition, fairness also involves a balanced overview of search results that represents different perspectives on a given topic [14]. Ensuring fairness may require ranking search results to present diverse perspectives in a just and equitable manner [23]. By combining diversity and fairness principles, IR systems can offer users more inclusive, unbiased search experiences.

The diversity becomes more important in multiple-query sessions, where users can interact with different perspectives and subtopics subsequently [53, 54]. In these sessions, a search topic often comprises multiple subtopics or diverse viewpoints, making it essential to promote result diversity. By encouraging exploratory search behaviors through multiple queries, users are potentially exposed to a varied range of information, facilitating more balanced or polarized opinions [7, 15, 28, 29]. Multi-query sessions also provide complex conditions for user acceptance of diverse content [39].

By directly or indirectly manipulating the search result diversity, various strategies have been proposed to mitigate the strength of users' opinions and reduce confirmation bias. Rieger et al. [55]

conducted an experiment on participants with moderate to strong preexisting attitudes toward debated topics. They utilized a result obfuscation on the SERP interface and reduced participants' clicks on attitude-confirming results. Draws et al. [17] and Gao and Shah [23] proposed diverse-viewpoint and fairness-aware ranking algorithms that take into account the diversity of search results to create a fairer ranking in search engine results. Pothirattanachaikul et al. [52] found that the "People Also Ask" feature can influence users' search behaviors and perceptions of search results, depending on the user's prior knowledge and attitudes toward the topic.

## 3 RESEARCH QUESTIONS

It is crucial to understand and mitigate confirmation bias in IR to avoid negative consequences such as reinforcing polarized biases and perpetuating harmful stereotypes. Presenting diverse perspectives and using unbiased algorithms are potential mitigation strategies. However, there are still gaps in the current research about users' opinions or attitude changes on debated topics and confirmation bias at the session level. To address this, we designed a three-query session on a simulated search interface about debated topics under various *SERP bias conditions* and conducted a *between-subjects* study with crowdsourcing participants. Specifically, this study aims to answer these research questions (RQs):

**RQ1** : How are users' within-session attitude changes associated with query-wise attitude changes and user characteristics?

**RQ2** : How does confirmation bias affect search interactions under SERP bias conditions?

**RQ3** : How do bias level and sequence position of biased SERPs affect users' within-session attitude changes?

## 4 METHODOLOGY

### 4.1 Crowdsourcing Study Design

Figure 1 shows the crowdsourcing study design, procedure, and interfaces. We designed a three-query search session with combinations of various biased SERP conditions (Tables 1 and 2) to allow crowdsourced participants to interact with the search interfaces under different conditions in a *between-subject* experiment.

*4.1.1 Task description.* For the task design, we adopted a method from previous studies about searching debated topics. The topics and opinionated documents are from two public datasets[1] collected by Draws et al. [18] and Rieger et al. [55]. The topics include: (1) *Are social networking sites good for our society?* (2) *Should zoos exist?* (3) *Is homework beneficial?* (4) *Should students have to wear school uniforms?* For each topic, the dataset contains relevant documents with a viewpoint score on a 7-point Likert scale from -3 to +3, representing the **direction** (+: supporting, or -: opposing) and the **strength** (1: somewhat, 2: normally, 3: strongly) towards debated topics. These topics were chosen based on their relevance to current societal debates and their potential to provide valuable insights from various perspectives. They offer sufficient information availability, ensuring that reliable sources of data can be accessed without undue difficulty. Given the importance of providing unbiased and reliable results, we deliberately avoided involving more controversial subjects related to politics, gender, race, or religion.

Such controversial topics have a higher likelihood of being emotionally charged and might raise ethical concerns due to the sensitive nature of these topics. By focusing on less controversial subjects, we sought to maintain a balanced and fair approach to the research.

*4.1.2 Study procedure and data collection.* In the crowdsourcing procedure, participants were introduced to a scenario about searching for a debated topic. They provided demographic information and characteristics and were randomly assigned one topic that they had opinions on. The user characteristics include preexisting attitude (same scale as the document viewpoint from -3 to +3), perceived prior knowledge level and perceived openness (both self-reported and measured on 7-point Likert scales), and information sources (5-category). The perceived openness is a degree to which participants agree to understand, respect, and be comfortable with a conflicting perspective when debating with others, which can reflect several key aspects of personality traits, such as openness to experience and conflict resolution skills [11]. The information sources include News, Personal experience, Personal conversation, Expert information, and Online non-expert information.

Participants who were not opinionated (i.e., 0: neutral attitude) on all four topics were unqualified for this study. Qualified participants were then randomly assigned to one session condition and interacted with the simulated search interface. They first selected a query term from a pre-defined query set, which is from previous studies [18, 55] and contains 14 queries in a random order. The query selection would not affect the search results but can reflect user search intents and information needs. When interacting with the SERP, participants could click, browse, and bookmark results as they do in a normal search engine. For each clicked result, they assessed the usefulness and credibility (4-point scale). In each query, they were required to click at least three unique results to ensure sufficient user engagement. After each query, participants reevaluated their attitudes towards the assigned topic and answered questions about the perceived diversity of and familiarity with clicked results (7-point Likert scale). Then, they proceeded to the next query until completing three queries. After the three queries, they reviewed the clicked results and chose explanations (6-category). Explanations for clicks include curiosity, personal interest, agreement, disagreement, debate, and clicking by accident.

*4.1.3 Participants.* We conducted this study on the online survey platform Qualtrics[2], incorporating a commitment request (i.e., a pre-study question asking participants to provide thoughtful answers) and an attention check during the study (a question required a specific option) to ensure data quality[3]. We recruited participants from Amazon MTurk[4] with requirements: at least 18 years old and native English speakers. We compensated them $2 for each 15-minute task. This study was approved by the Institutional Review Board. As a result, 1,500 participants were recruited, and 1,321 who passed quality checks were analyzed. The participants include 906 males, 410 females, and 5 not specified, with an average age of 34. Participants initially tended to have a supporting attitude to the topic (mean=1.9±1.2) and perceived good or relatively good knowledge and openness of the topic (mean=5.9±0.9; mean=5.9±0.8). The

---

[1]https://osf.io/6tbvw/; https://osf.io/32wym/

[2]https://www.qualtrics.com/

[3]https://www.qualtrics.com/blog/attention-checks-and-data-quality/

[4]https://www.mturk.com/

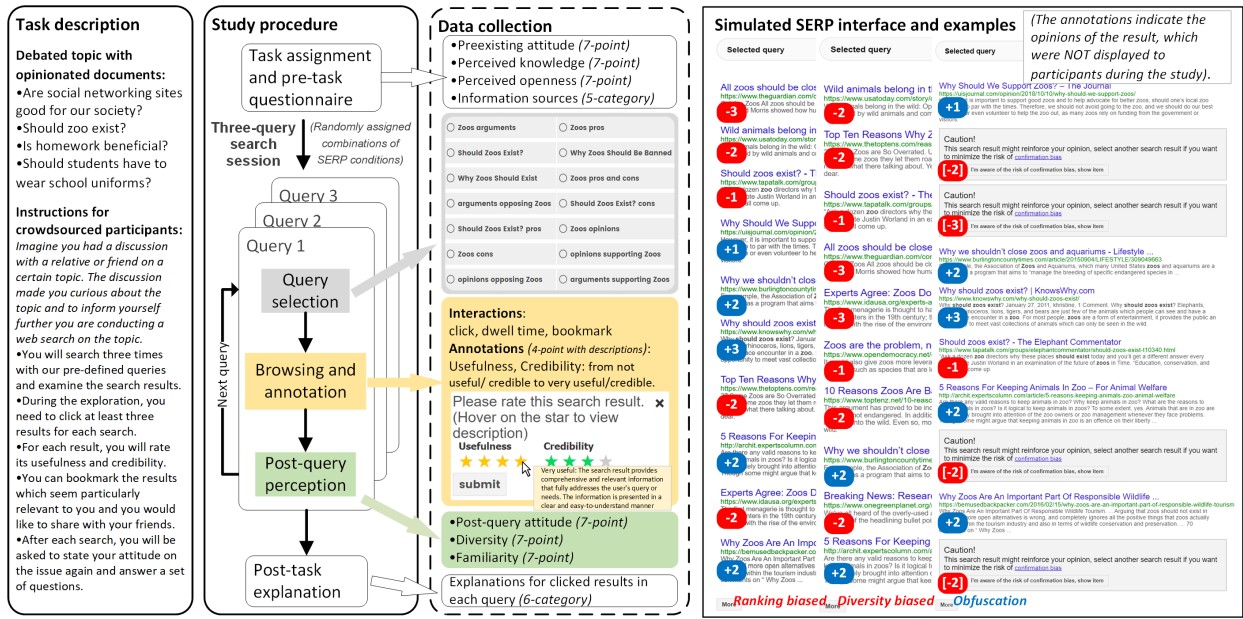

**Figure 1: Study procedure and interfaces.**

participants also indicated multiple information sources they utilized to obtain information about the topic before taking the survey. The most common sources were personal experience (n=707), news (n=706), and expert information (n=610), followed by personal conversation with others (n=448) and online non-expert information (n=240).

*4.1.4 SERP and session conditions.* We first designed biased SERP conditions with opinionated documents and investigated the effects on search interactions. The SERP conditions designed in this study can reflect biased situations in real search scenarios by simulating different types of biases that search engines might produce, such as the ranking bias and the diversity bias (SERP examples illustrated in the right side of Figure 1), distinguished by the statistical (im)parity [23] to represent conditions when the main bias is from the positions or amount of attitude-confirming results.

Specifically, we organized the opinionated documents under the biased SERP conditions in Table 1, inspired by previous studies that focused on ranking bias [18, 55]. Beyond that, we also designed diversity-biased conditions and their counterparts as mitigated conditions. According to Table 1, the SERP consists of ten (6+4) results. We changed the ranking or constitution of the top six documents to represent different bias levels and keep a fixed order of the rest four as we expected users to mostly interact with the top results. The ranking orders of the balanced (**Bal**) and the ranking biased (**R**) conditions are based on a fairness score calculated by the sum of three metrics: normalized discounted difference (nDD), normalized discounted Kullback-Leibler divergence (nDKL), and normalized ranking bias (nRB) [17, 30], so **Bal** has the lowest bias, and **R** has the highest bias in all ranking permutations. In the diversity-biased (**D**) condition, the top six results are all holding the same opinions. Mitigated conditions are counterparts of the ranking and diversity bias conditions (**Counter-R** and **-D**) and targeted obfuscation (**Obf**,

**Table 1: SERP conditions with opinionated documents.**

| Rank | Biased | | Bal- | Mitigated | | |
|---|---|---|---|---|---|---|
| | R(ank) | D(iversity) | (anced) | Counter-R | Counter-D | Obf(uscation) |
| 1 | -3 | -2 | -1 | +3 | +2 | +1 |
| 2 | -2 | -2 | +2 | +2 | +2 | [-2] |
| 3 | -1 | -1 | +3 | +1 | +1 | [-3] |
| 4 | +1 | -3 | -2 | -1 | +3 | +2 |
| 5 | +2 | -3 | -3 | -2 | +3 | +3 |
| 6 | +3 | -1 | +1 | -3 | +1 | -1 |
| 7 | -2 | -2 | -2 | +2 | +2 | +2 |
| 8 | +2 | +2 | +2 | -2 | -2 | [-2] |
| 9 | -2 | -2 | -2 | +2 | +2 | +2 |
| 10 | +2 | +2 | +2 | -2 | -2 | [-2] |

The negative numbers represent results with opposing opinions, and the positive numbers represent supporting opinions. The [numbers in brackets] represent the obfuscated results. This table show conditions for users with opposing attitudes, and we have also included conditions for users with supporting attitudes in this study.

also illustrated in the right side of Figure 1), which prioritize alternative opinions or obfuscates results with strong opinions to avoid reinforcing users' existing bias [6, 52, 55, 70]. In addition, there is a "More" button at the bottom of the SERP which will unhide three results with a neutral opinion for participants who want to browse more results beyond the ten opinionated results.

To represent scenarios that users might experience in a three-query session, we designed session conditions with different *bias levels* and *positions* (Table 2). The bias level represents the number of biased query(-ies), including "No query biased", "One query biased", "Two queries biased", "One query mitigated", and "Mitigated and biased". Each bias level condition contains two or three sub-conditions with biased query(-ies) at different positions.

## 4.2 Investigating Characteristics of Within-session Attitude Changes

From the pre-task questionnaire and the three-query session, we collected self-reported attitudes at four points: preexisting attitude

**Table 2: Five main session conditions with different bias levels, each with sub-conditions of different bias positions.**

| Level | Position | SERP conditions | | |
|---|---|---|---|---|
| | | **Query 1** | **Query 2** | **Query 3** |
| No bias | \ | Bal | Bal | Bal |
| One query biased | First | R/D | Bal | Bal |
| | Second | Bal | R/D | Bal |
| | Third | Bal | Bal | R/D |
| Two queries biased | First&second | R/D | R/D | Bal |
| | Second&third | Bal | R/D | R/D |
| One query mitigated | First | Counter/Obf | Bal | Bal |
| | Third | Bal | Bal | Counter/Obf |
| Mitigated and biased | First | Counter/Obf | R/D | Bal |
| | Third | Bal | R/D | Counter/Obf |

*: Bal: balanced condition; R/D: ranking or diversity bias; Counter: mitigation with counterpart of ranking or diversity bias; Obf: targeted obfuscation.

(**Q0**) and attitudes after each query (**Q1/2/3**), and calculated query-wise attitude changes between two consecutive queries (**Qn+1 - Qn**). Furthermore, we calculated two within-session attitude changes, including: *accumulative attitude change*, defined as the sum of absolute query-wise attitude changes, reflecting the total fluctuation of users' attitude; and the *directional attitude change*, defined as the differences between the preexisting attitude and the post-session attitude in the direction of preexisting attitude (**(Q3 - Q0)*direction**), representing the overall reinforcement (or mitigation) on attitude strength. We used "attitude changes" to denote the two within-session attitude changes unless otherwise specified.

To address **RQ1**, we explored the characteristics of attitude changes via descriptive analysis. First, we compared query-wise and within-session attitude changes. Then, we investigated the correlations between user characteristics and attitude changes.

## 4.3 Investigating Effects of Confirmation and SERP Biases on Search Interactions

For **RQ2**, we analyzed how the confirmation bias at the session level affects search interactions. These interactions (Table 3) are grouped into click-based, time-based, perception-based, and other features. Then, we investigated the effects by analyzing the differences in search interactions among SERP conditions. Under the three-query session, these effects can be investigated at the *within-query* level: the immediate effects on interactions in the current query, and the *between-query* level: extended effects of the previous query on interactions in subsequent query.

For effects at the *within-query* level, we used the SERP conditions presented in Table 1 as the independent variable and analyzed the differences in search interactions in Table 3 across the SERP conditions. Specifically, we formulated hypotheses:

**H1a**: Users tend to click more results or explore results at lower ranks (i.e., higher ClickNum and ClickDepth) in the mitigated SERP conditions compared to other conditions during the current query.
Rationale: As the SERPs in the mitigated conditions prioritize attitude-disconfirming results, we expect that the confirmation bias increases users' clicks for exploring more attitude-confirming results.

**H1b**: Users tend to spend less time on average per result (i.e., lower TimeAvg) in the mitigated SERP conditions than other conditions during the current query.

**Table 3: Search interaction features.**

| Feature | Description |
|---|---|
| *Click-based features* | |
| **ClickNum** | Number of clicks in the query. |
| **ClickDepth** | The lowest rank of clicked results. |
| ClickRank | Average rank of clicked results. |
| ClickProp | Ratio of clicks on attitude-confirming results to all clicks *(Normalized by two times of percentage of attitude-confirming results in the SERP).* |
| *Time-based features* | |
| **TimeAvg** | Average dwell time of clicked results (Second). |
| TimeProp | Ratio of average dwell time of attitude-confirming results to all clicked results. |
| *Perception-based features* | |
| **UseAvg** | Average usefulness score of clicked results. |
| **CredAvg** | Average credibility score of clicked results. |
| **Diversity** | Users' perceived diversity of clicked results. |
| **Familiarity** | Users' perceived familiarity with clicked results. |
| UseProp | Ratio of average usefulness of attitude-confirming results to all annotated results. |
| CredProp | Ratio of average credibility of attitude-confirming results to all annotated results. |
| *Other features* | |
| NextQuery | Preference of the next query (positive/negative/neutral). |
| MarkAvg | Average bookmark rate of clicked results. |
| ClickMore | The ratio of users who clicked the "More" button. |

Features in **boldface** indicate dependent variables. Others are descriptive variables.

Rationale: We expect that the users spend less time on both attitude-disconfirming and -confirming results because of the effects of confirmation bias and bias mitigation.

**H1c**: Users tend to have different perceptions of the results (e.g., UseAvg, CredAvg, Diversity, Familiarity) among the SERP conditions of the current query.
Rationale: We expect users to assess or perceive the attitude-confirming and -disconfirming results differently.

For H1a-c, we examined the features across SERP conditions at the first query to avoid potential effects from previous queries. Besides the dependent variable features, we also considered other search interactions as descriptive variables, including the feature variants (e.g., ClickRank, UseProp) and other behavioral features (e.g., NextQuery) to enhance our analysis and support findings from hypothesis testing, providing a comprehensive view of the confirmation bias effects.

For the *between-query* effect, we used the session conditions in Table 2 as the independent variable and formulated the hypothesis:
**H2**: Users tend to have different search interactions (i.e., click-, time-, or perception-based features) in the query if they encountered different SERP conditions in previous queries in the session.
Rationale: We expect that the confirmation bias makes users change their search strategies if they previously engaged with different amounts of attitude-confirming and -disconfirming results.

For H2, we examined search interactions at the second or third query with a balanced condition to control the current query's SERP condition. For the statistical tests in H1, H2, and descriptive variables, we implemented the Kruskal-Wallis test as the non-parametric test to investigate the differences in each feature among these conditions. We utilized the Benjamini–Hochberg method to

control the false discovery rate for testing multiple search interaction features. For features with significant differences, we then utilized the Conover-Iman test as the post hoc pairwise test to determine which groups have significant differences [12].

### 4.4 Effects of Bias Level and Position on Within-session Attitude Changes

For **RQ3**, we investigated the differences in attitude changes across session conditions. We used the five main bias level conditions and the bias position sub-conditions as the independent variables and attitude changes as the dependent variables. Our hypotheses are:

**H3a** : Users' tend to have different attitude changes when they encounter different bias levels.

**H3b** : Users' tend to have different attitude changes when they encounter the biased query at different positions in a session.

For H3a, we analyzed if there are differences in the attitude changes among the five bias level conditions. For H3b, we analyzed if there are differences in the attitude changes among sub-conditions of bias positions for each bias level condition. Furthermore, we also investigated the confounding effect between user characteristics and session conditions by controlling the use group. We implemented the Kruskal-Wallis test to examine the differences in attitude changes among session conditions (as the data was not normally distributed across groups) and utilized the Benjamini–Hochberg correction method to control the false discovery rate [5].

## 5 RESULTS

### 5.1 Within-session Attitude Changes

To answer RQ1, we used descriptive analysis to explore the within-session attitude changes. On average, participants in this study experienced an accumulative change of 1.59 and a directional change of -0.82, indicating a minor fluctuation and a slight mitigation from an extreme attitude to a neutral attitude. Table 4 presents the components of directional and accumulative changes, which are linearly composed of either directional or absolute query-wise changes. In general, the attitude change after the first query contributes most to the directional change and accounts for about half of the accumulative change. Although the average change after the second or the third query is close to zero, the large standard deviation indicates that participants experienced either reinforced or mitigated attitude strength. Thus, the absolute values of those changes are not zero, contributing to a portion of the accumulative change. We then analyzed the correlations between attitude changes and user characteristics. Table 5 presents the correlation matrix of Spearman's Rho. According to the results, participants with stronger preexisting attitudes tended to show more significant accumulative and mitigated changes. In addition, although attitude strength is positively correlated with perceived knowledge and openness, participants with higher perceived knowledge or openness showed less accumulative attitude change. We also did analysis within individual task topics respectively and obtained similar results.

In summary, these results underscore the role of the first query in shaping directional attitude changes, and the second and third queries still contribute to the accumulative change during the session. The within-session attitude changes are also influenced by user

**Table 4: Mean (±SD) of attitude changes.**

| Change | Q1-Q0 | Q2-Q1 | Q3-Q2 | Directional | Accumulative |
|---|---|---|---|---|---|
| Mean | -0.76±0.85 | -0.08±0.77 | 0.03±0.81 | -0.81±0.98 | - |
| Absolute | 0.81±0.80 | 0.38±0.67 | 0.40±0.70 | - | 1.59±1.56 |

**Table 5: Attitude changes and user characteristics.**

| Spearman's Rho | AttStrength | Knowledge | Openness | Directional |
|---|---|---|---|---|
| Knowledge | 0.54** | - | - | - |
| Openness | 0.37** | 0.39** | - | - |
| Directional | -0.28** | -0.02 | -0.04 | - |
| Accumulative | 0.10** | -0.10** | -0.07* | -0.53** |

*: $p<0.5$, **$p<0.1$

characteristics. Notably, some results, particularly regarding the negative relationships between accumulative changes and preexisting attitude strength & perceived openness, were counter-intuitive, prompting further exploration in our discussion section.

### 5.2 Effects of Confirmation Bias on Search

To answer RQ2, We examined the within- and between-query effects of SERP conditions on search interactions. Table 6 shows the average values of interactions across SERP conditions in the first query, reflecting the within-query effect. Generally, the results show that there are significant differences mainly in click-based interactions among different SERP conditions. The mitigated conditions of the counterpart of ranking bias (**Counter-R**) and targeted obfuscation (**Obf**), differ significantly in these features from other conditions. Specifically, in these two conditions, users had more clicks and/or clicks at lower ranks compared to other conditions, probably for seeking more attitude-confirming results. The normalized ClickProp is different among all conditions, where the condition of ranking bias (**R**) has the highest, and the condition of targeted obfuscation has the lowest. These results indicate that manipulating the SERP presentation can lead users to results with different opinions. In addition, interactions with the attitude-confirming results at lower ranks in the Counter-R condition can be an indicator of confirmation biases, since the user could locate those results from the title and snippet. Therefore, we accepted hypothesis H1a that users tend to click more results in the Counter-R condition and explore results at lower ranks in the Counter-R and Obf conditions.

For time-based interactions, users would spend less time in attitude-confirming results in the conditions of Counter-R and Obf than users in other conditions, which can be caused by the effect of manipulating the SERP presentation. However, there was no difference in average spending time. Therefore, we rejected hypothesis H1b and accepted that users tend to spend a similar amount of time on average per result in all SERP conditions during the current query. In addition, the perception-based interactions do not exhibit any significant differences among the SERP conditions. This suggests we reject hypothesis H1c and accept that the SERP condition settings and confirmation bias did not impact users' assessments or perceptions of the search results in the current query.

Furthermore, confirmation bias could be observed when comparing the rates of participants who clicked both attitude-confirming and disconfirming results between the conditions of ranking bias

**Table 6: Search features across different SERP conditions in the first (current) query. Bal: balanced condition; R/D: ranking or diversity bias; Counter-R/D: mitigation with counterpart of ranking or diversity bias; Obf: targeted obfuscation.**

| SERP | Bal | R | D | Counter-R | Counter-D | Obf |
|---|---|---|---|---|---|---|
| Count | 138 | 133 | 131 | 131 | 132 | 128 |
| Rate[1] | 0.93 | 0.67 | 0.27 | 0.91 | 0.27 | 0.62 |
| *Click-based features* | | | | | | |
| **ClickNum**** | 3.97 | 3.80 | 3.62 | 5.30 | 3.55 | 3.54 |
| **ClickDepth**** | 7.17 | 6.77 | 6.58 | 8.48 | 6.37 | 8.12 |
| ClickRank** | 4.53 | 4.46 | 4.35 | 5.02 | 3.96 | 5.71 |
| ClickProp** | 0.50 | 0.65 | 0.53 | 0.48 | 0.22 | 0.21 |
| *Time-based features* | | | | | | |
| **TimeAvg** | 11.62 | 10.54 | 15.4 | 13.76 | 11.63 | 11.02 |
| TimeProp** | 1.00 | 1.07 | 1.04 | 0.92 | 0.94 | 0.93 |
| *Perception-based features* | | | | | | |
| **UseAvg** | 2.83 | 2.98 | 2.91 | 3.00 | 2.91 | 2.94 |
| **CredAvg** | 2.98 | 3.04 | 3.05 | 3.03 | 3.04 | 3.02 |
| **Diversity** | 5.21 | 5.39 | 5.37 | 5.27 | 5.49 | 5.22 |
| **Familiarity** | 5.21 | 5.35 | 5.39 | 5.10 | 5.46 | 5.04 |
| UseProp | 1.01 | 1.00 | 1.01 | 0.99 | 1.00 | 0.99 |
| CredProp | 1.01 | 1.01 | 1.01 | 1.00 | 0.97 | 1.04 |
| *Other features* | | | | | | |
| NextQuery | 0.05 | 0.11 | 0.08 | 0.04 | 0.11 | 0.09 |
| MarkAvg | 0.37 | 0.40 | 0.45 | 0.41 | 0.47 | 0.46 |
| ClickMore* | 0.23 | 0.14 | 0.15 | 0.27 | 0.18 | 0.27 |

[1] The rate of participants who clicked both attitude-confirming and disconfirming results.
Features in **boldface** indicate dependent variables. Others are descriptive variables.
* indicates significant differences across SERP conditions: * corrected $p < 0.05$, ** corrected $p \ll 0.01$. The value with underscore indicates significant differences between the condition and two or three other conditions with corrected $p < 0.05$.

(67%) and counterpart of ranking bias (91%), in which they might tend to find more attitude-confirming results at lower ranks if reading attitude-disconfirming results at top ranks. The ClickMore rate also suggests users in these two conditions (27%) wanted to explore more results than users in other conditions.

Beyond the within-query effects, we also investigated the between-query effect, exploring how these conditions in earlier queries might influence user interactions in subsequent queries. We found that the perceived familiarity with clicked results in the third query exhibited significant differences when participants experienced different conditions in the first and second queries (see Figure 2). Specifically, participants reported lower perceived familiarity in the third query if they were presented with obfuscation in the first query and a balanced SERP in the second query. Conversely, higher perceived familiarity was mainly reported if participants experienced ranking-biased conditions in the second query. Thus, we partially accepted hypothesis H2 that users' perceived familiarity with clicked results is likely to be different if they previously encountered SERPs with different conditions, indicating that participants under these conditions could be exposed to more (or less) familiar results.

## 5.3 Effects of Bias Level and Position on Witin-session Attitude changes

We compared the attitude changes among session conditions to investigate the effects of bias levels and positions. However, we did not observe significant differences in accumulative or directional change among the five conditions of bias levels or among sub-conditions of bias positions. This indicates that the session conditions did not lead to significant variations in the impact of confirmation bias on attitude changes.

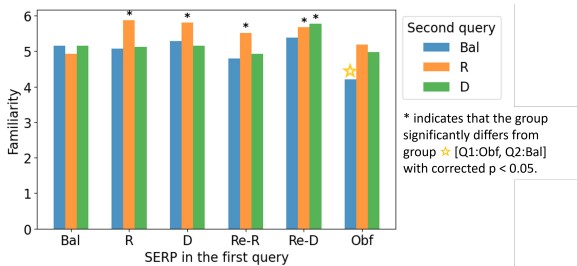

**Figure 2: Perceived familiarity of the third query with balanced SERPs ([Q1: Obf, Q2: Bal] represents the condition where the first query has a SERP with obfuscation, and the second query has a balanced SERP).**

**Table 7: Directional attitude changes in the user group with lower openness (<6) across conditions of bias positions.**

| Bias level | Bias position | Count | Q1-Q0 | Q2-Q1 | Q3-Q2 | Directional change |
|---|---|---|---|---|---|---|
| One query biased | First | 30 | -0.83 | -0.07 | -0.37 | -1.27 |
| | Second | 38 | -0.71 | -0.03 | 0.24 | -0.50 |
| | Third | 19 | -0.74 | -0.11 | -0.05 | -0.89 |
| Two queries biased | First&second | 39 | -0.74 | 0.28 | -0.13 | -0.59 |
| | Second&third | 46 | -0.96 | -0.09 | -0.02 | -1.07 |

The directional changes are significantly different across bias positions with corrected $p < 0.05$.

We then investigated the confounding effect of user characteristics and session conditions on attitude changes. We divided participants into groups based on the median values of preexisting attitude strength, perceived knowledge, and perceived openness, separately, and tested the differences in attitude changes among five conditions of bias levels or among sub-conditions of bias positions. We found significant differences in directional attitude changes across conditions of bias positions within the user group that had perceived openness lower than the median (<6). Table 7 presents the different directional changes with query-wise change portions. The results indicate that attitude strength was slightly mitigated after the first query regardless of the SERP condition but could be reinforced after the second or third query with the first and/or second queries biased. This finding indicates that users with lower perceived openness are more susceptible to directional attitude changes when influenced by the sequence position of biased SERPs.

However, it is important to note that we did not control for user characteristics during the recruiting process, leading to imbalanced user groups in certain conditions. These imbalanced data may result in inaccurate estimations and comparisons. Nevertheless, as we recruited participants and assigned task conditions independently, the results revealed a confounding effect of the bias positions and users who reported lower perceived openness. Therefore, we partially accepted H3b, suggesting that users with lower perceived openness are likely to have different directional attitude changes if encountering biases at different query positions in the session.

## 5.4 Evidence from Case Studies

To extract evidence to support and explain the above findings, we delved deeper into individual participants' behaviors as case studies based on user characteristics, query selection, clicks, attitude

change, and post-task explanations. We selected example participants to further illustrate four distinct situations: attitude strength was mitigated, reinforced, unchanged, or fluctuated.

*Mitigated:* Participant 203 (P203) held a strongly supporting attitude (+3) toward zoos, with rather good knowledge (5) and high openness (6), and their (a gender-neutral pronouns) information source was news. In the first query, P203 first clicked on the top results (attitude-disconfirming) and then three attitude-confirming results. Then, their attitude strength was mitigated to somewhat supporting (+1). In the second and third queries, P203 first clicked attitude-disconfirming results at the bottom and then returned to results at higher ranks. Their attitude became neutral (0) and then somewhat opposing (-1). The explanations for clicks include personal interest and curiosity.

*Reinforced:* P578 started with an opposing attitude (-2) on social networks, with fair knowledge (4) and moderate openness (4), and the information sources include personal experience and conversations. Throughout all three queries, P578 consistently clicked attitude-confirming results regardless of rank. After the first query, P578's attitude strength was reinforced to strongly opposing (-3). The clicks were explained by personal interest, curiosity, and agreement with the results' opinions.

*Unchanged:* With the initially strong support (+3) on homework, the highest levels (7) of knowledge and openness, and diverse information sources, P904 consistently supported homework across all queries, unaffected by results of either viewpoint at any ranks, suggesting a robust belief. The complex explanations of clicks include personal interest, curiosity, agreement, disagreement, and debate.

*Fluctuated:* P1300 had a somewhat supporting (+1) view on school uniforms, with limited knowledge (2) and openness (1). The information sources were diverse. P1300 clicked all attitude-confirming results in the first query, and their attitude changed to somewhat opposing (-1). Then in the second and third queries, P1300 clicked explore both attitude-confirming and -disconfirming results, but their attitude changed back to somewhat supporting (+1) and ended at opposing (-2). The main explanations are personal interest, curiosity, and agreement.

## 6 DISCUSSION

In this study, We analyzed users' interactions under confirmation bias and SERP bias in three aspects: the characteristics of attitude changes, the effects of confirmation bias on search interactions, and the effects of bias level and positions on attitude changes. We tested hypotheses about the effects of bias conditions, and the results are summarized in Table 8. We further explored how these findings enhance our understanding of the RQs and discussed possible explanations based on the evidence from the case study.

### 6.1 RQ1. Within-session Attitude Changes

To answer RQ1, we examined directional and accumulative attitude changes in relation to query-wise changes and user characteristics. Our results reveal that the majority of attitude changes occur in the first query. Preexisting attitude strength positively correlates with accumulative changes and negatively with directional changes, while perceived prior knowledge and openness to conflicting opinions show negative correlations with accumulative changes.

**Table 8: Summary of hypothesis testing results under RQ2.**

| | Hypothesis result |
|---|---|
| H1a | **Accepted:** Users tend to click more results in the Counter-R condition and explore results at lower ranks in the Counter-R and Obf conditions. |
| H1b | **Rejected:** Users tend to spend a similar amount of time on average per result in all SERP conditions during the current query. |
| H1c | **Rejected:** The SERP condition settings did not impact users' assessments or perceptions of the search results in the current query. |
| H2 | **Partially accepted:** Users' perceived familiarity with clicked results tend to be different if they encountered SERPs with different conditions in the previous queries |
| H3a | **Rejected:** The bias level conditions did not affect users' attitude changes. |
| H3b | **Partially accepted:** Users with lower perceived openness tend to have different directional attitude changes if encountering biases at different query positions in the session. |

Regarding the most attitude changes in the first query, it is possible that the participants perceived new information with a primacy effect during the first query, which might have more impact on their attitudes and search behaviors than subsequent queries [21, 43]. Participants might be exposed to new information in the first query (e.g., P203 with simple information sources). After the first query, participants might reach a saturation point due to information overload [58]. With other potential inherent biases in user study (e.g., Hawthorne effect [1] and demand characteristics [45]), participants might assume that the study expects them to show attitude change and be ready to adjust their opinions regardless of conditions. This bias may not cause differences in attitude changes across conditions if the search activities occur in traditional ad hoc retrieval [18].

Regarding user characteristics, it is reasonable that users with higher perceived levels of knowledge are likely to have more stable attitudes and opinions, probably because they have a more established understanding of the topic (e.g., P904). However, for the counterintuitive observations, users with stronger but not robust initial attitudes experienced greater changes and mitigation probably because they have more room for mitigation and *polarization reduction.* Strong or extreme attitudes might naturally become less extreme when exposed to information with alternative perspectives (e.g., P203) [4]. As the perceived knowledge and openness to conflicting opinions are positively correlated, users with higher perceived openness may already be familiar with diverse perspectives and arguments of the debated topics, making their attitudes more stable. In addition, they may scrutinize new information more thoroughly before accepting it or be curious about the new perspective to prepare for debates (e.g., P904's explanations). On the contrary, users perceived lower knowledge may have low openness but an unestablished attitude (e.g., P1300). Furthermore, as we used a single statement to represent this complex construct, users may perceive this statement and openness differently. Thus, a more comprehensive survey may allow us to measure different facets of openness to diverse perspectives more accurately.

### 6.2 RQ2. Confirmation Bias Effects on Search

To answer RQ2, we investigated the effect of confirmation bias on search interactions. We observed differences in the number and depth of clicks across SERP bias conditions in the current query. The SERP presentation could also affect users' perceived familiarity with results in subsequent queries. The differences in click-based behaviors could be influenced both by SERP presentation and by

confirmation bias. On one hand, certain presentations (e.g., the counterpart of ranking biased SERP and obfuscation) prioritize results with alternative opinions, leading users to engage with those results first. On the other hand, these presentations still provide an equal amount of attitude-confirming results, and users increase their engagement with results even at lower ranks because of confirmation bias. However, interventions involving obfuscation or the counterpart of diversity bias might backfire, reducing user engagement due to *cognitive dissonance* [33, 50, 56, 70]. In addition, users encountering ranking-biased SERPs also demonstrated potential confirmation bias in terms of perceived familiarity in subsequent queries. Their higher familiarity might be caused by clicking more attitude-confirmation results (e.g., P578 clicked results for agreement). Conversely, if users initially encountered an obfuscated SERP and then shifted to a non-biased (i.e., balanced) SERP, they might not have been exposed to familiar results as extensively as users in other conditions. They could also be more adaptable to unfamiliar results in subsequent queries.

### 6.3 RQ3. Bias Positions Affecting Directional Attitude Changes

For RQ3, we found that users with lower openness to conflicting opinions could be influenced by the sequence position of biased SERPs. If this openness represents users' knowledge or previous experience in coping with diverse perspectives, the lower openness could mean that these users are more sensitive to hidden biases instead of being reluctant, leading them to react strongly when confronted with biased opinions (e.g., P1300's unstable attitudes). When bias occurs early in the search session (e.g., bias at the first query), it could evoke an immediate and intense reaction, resulting in a more significant directional change towards mitigating their initial extreme attitudes. On the other hand, encountering biased results later in the session (e.g., biased SERPs at the second or third query) might lead these users to a more thoughtful change in attitudes. This sequence effect of biased SERPs suggests that the confirmation bias could be confounded with *reference-dependent* effect in multi-query search sessions [3, 9, 38].

Additionally, the patterns of attitude change observed in the case study indicate further research on different user groups based on whether they were affected by the bias conditions. Despite the manipulation of SERPs, some users selectively reviewed results that confirmed their existing attitudes without influence from the result ranking (e.g., P578), while others engaged with opposing viewpoints with unchanged attitudes (e.g., P904). Such interactions could cause the results of no significant differences in attitude changes across bias conditions. However, greater attention should be given to those participants whose attitudes were swayed by the biased conditions. Further research is needed to identify users' characteristics that make their attitudes more sensitive.

### 6.4 Implications and Limitations

As Rieger et al. [55, 56] suggested, the *Elaboration Likelihood Model* [cf. 49], which differentiates between the *peripheral route* (e.g., manipulation effects of changing result presentations) and the *central route* (e.g., guiding users in the reflective process) in influencing user attitudes, can also explain results in our study. The results

in users' attitude changes and search interactions observed in the first query primarily reflect the peripheral route, which is consistent with previous research on single-query sessions [55, 56]. However, as users progress to later queries, a shift towards the central route may occur. This shift is evidenced by the influence of the sequence position of biased queries (in certain user groups), suggesting deeper cognitive processing and dynamic engagement in subsequent queries. Thus, this study highlights the importance of investigating user interactions in multi-query sessions under the impact of biases from both human and system sides, especially considering user characteristics and bias positions, and helps complement mainstream research focusing on algorithmic biases and simulation-based evaluations. The findings also raise important ethical considerations regarding the potential risks of *algorithmic manipulation* in human-information interactions [13]. To avoid misuse of these techniques for profit-driven search optimizations and deceptive purposes, intelligent safeguards need to be put in place by search developers, researchers, and AI policymakers. Furthermore, our findings can also inform the development of guidelines on building unbiased IIR systems and ethical AI audit techniques [60].

Our study has several limitations, including employing a limited set of topics, potential variations in participant populations, and utilizing self-reported values in measuring user perceptions. These limitations call for future efforts to continue this line of research on the interplay of cognitive biases, algorithmic biases, and search contexts. To mitigate the impact of limitations, our study employs a wide range of topics that are diverse and more appropriate and accessible as common debatable topics than other controversial topics. With self-reported characteristics, our work revealed the effect of confirmation bias in certain user groups and paved the way for future experiments on mixed bias effects in search sessions. Furthermore, the case study showed the diverse patterns of user search interactions and attitude changes. In spite of its limitations, the study contributes to our understanding of the confirmation bias in multi-query sessions and demonstrates a viable approach to examining task-driven interactions between cognitively biased users and algorithmically biased SERPs under reasonably authentic experimental settings. Future studies can explore a broader range of high-stake topics (e.g. medical decisions, personal health management, financial investments), investigate the long-term effects of biased results and system-generated contents on attitude change in natural environments, and examine the role of social contexts and neural correlates of biased search behaviors. Knowledge about biases can also help reduce the gap between IR evaluation measurements and users' *in-situ* judgments and experiences [9, 37].

## 7 CONCLUSION

In conclusion, this study provides insight into the interaction between users, affected by confirmation bias, and algorithmically biased SERPs in whole-session searches on debated topics. It advanced our understanding of the long-term effects of confirmation bias on users' search interactions, judgments, and attitudes. Future studies can continue to investigate the complex effects of biases in a broader range of human-AI interaction scenarios, identify the hidden biases in retrieved information and system-generated responses, and deploy effective bias mitigation techniques.

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
