# OpenReview forum: "Cognitively Biased Users Interacting with Algorithmically Biased Results in Whole-Session Search on Debated Topics"
_ACM.org/SIGIR/ICTIR/2024/Conference — ICTIR 2024_

### Official Review · Reviewer_PTVc · 2024-05-15

**Rating:** 1
**Confidence:** 3

**Objective Part Of Review:**

In this paper, the authors investigate how different biases (cognitive and algorithmic) can impact user behavior and search outcomes. They frame this in the search for debated topics. The authors describe a user study with explorative and confirmatory results analyses. They conclude that users' attitude changes largely occur in the initial phase of search sessions, that users' confirmation biases can interact with algorithmic ranking biases to produce more biased behavior, and that interactions are related to users' attitude changes.

The study is well-motivated and an original piece of work that addresses a relevant and important problem. The literature review is good, with a very extensive reference list.

**Subjective Part Of Review:**

My two main concerns with this work are the rather high complexity of the study and its results, as well as the presentation and the general relation to ICTIR's focus on theoretical aspects of IR.

1) Complexity and presentation
- Tables reaching over the edge, too packed, too small print
- Highly complex study, I found myself lost here and there while reading the paper.
- Many participants are integrated in the study, but with little knowledge about them.
- Why so many interaction features (Table 3)?
- Tables 4 and 5 seem a bit misleading. There seems to be no significant difference in attitude changes.

2) Theoretical aspects
- Overall, I asked myself: In a nutshell, why is this an ICTIR submission and not a regular SIGIR or ECIR submission?
- While the paper is highly relevant to the IR community and addresses an important problem, I missed the theoretical center point/discussion a bit. Things like confirmation bias and exposure effect are not new and well-studied (also in IR). So, where is the (new) theoretical contribution to the field, or what open theoretical issue is tackled with a practical implementation/evaluation? I might have missed it, though.
- I see the mention of SEME (Epstein et al.), but this work has not really been picked up and discussed. Although I liked the extensive discussion in section 6, I missed a relation to Epstein's work.

Overall, I see some merits and an interesting orientation, but some potential for improvements.

---

### Official Review · Reviewer_fCLK · 2024-05-16

**Rating:** -2
**Confidence:** 4

**Objective Part Of Review:**

The paper studies the effect of cognitive biases of users in a search session where the search results themselves might also be biased.

**Subjective Part Of Review:**

The paper is nicely written and easy to follow. There are three research questions addressed in a user study.

However, in my opinion, this paper is not in the scope of ICTIR.

The CFP states the following:

In any case, the theoretical part should be a significant, integral, and non-trivial part of the paper, and not just an add-on. The term "theoretical" is meant in a comprehensive sense, encompassing not only mathematical work, but also conceptual work, all kinds of modelling (for example, models of user behavior), problem generalizations, etc.

There is no modeling part in the paper nor theory. There are three solid research questions address using a user study.
Hence, I don't see why this paper fits the scope of ICTIR.

---

### Official Review · Reviewer_fMxN · 2024-05-17

**Rating:** 0
**Confidence:** 3

**Objective Part Of Review:**

This paper studies the attitude change in multi-query search sessions under different bias conditions. The problem and experimental results are clearly described.

On the other hand, it is not very clear how the session conditions were designed and biased SERPs were presented to the user. Table 2 shows various session conditions, but their design principle is not well discussed (and may be problematic as pointed out later). The motivation of using obfuscates results is not fully discussed. If I understand correctly, they are not algorithmically biased results. Each bias level condition contains sub-conditions but there is no explanation on how they are used or assigned to the user (my guess is that the user was first assigned to one of the top-level conditions and then randomly assigned to one of its sub-conditions).

Some rationales given to each hypothesis are not clear enough: e.g., “We expect that the users spend less time on both attitude-disconfirming and -confirming results because of the effects of confirmation bias and bias mitigation.” The authors also claim that “the confirmation bias increases users’ clicks for exploring more attitude-confirming results.” A natural inference for the reading time is that “the confirmation bias increases the reading time for exploring more attitude-confirming results”, which contradicts the rationale. In addition, it is not clear why the bias mitigation is a part of the reason of less reading time.

The major concerns in the experimental design are:

1.	SERPs shown in response to the second query are different from those shown to the first and third queries. The user never observed Counter/Obf in the second SERP and had less chances to experience balanced SERPs. This problem makes the comparison of Q1-Q2 and Q2-Q3 difficult. Although the authors pointed out the unique effect of Q1, it is possible that the effect of Q2 is the same as that of Q1 if their SERP distributions are the same.
2.	“They first selected a query term from a pre-defined query set, which is from previous studies [18, 55] and contains 14 queries in a random order. The query selection would not affect the search results.” It is not clear whether this setting can simulate multi-query sessions. The first query from users could be a general query, but the second and third queries could be input with some intents (e.g., specification and generalization). However, they can only see the same set of queries and SERPs are not affected by the query selection. This situation is almost equivalent to the case where the user simply browses more SERPs by moving to the next SERP under the same query. In reality, users may input a new query to look for more opposing or supporting opinions. Therefore, I’m not very sure to what extent this study could simulate real multi-query sessions.

**Subjective Part Of Review:**

The paper is well structured and written. Most of the sections are easy to follow.

The results in Figure 2 are not well explained. In addition, the discussion on this issue in Section 6.2 is very hard to follow.
In particular, I still do not understand possible reasons for more/less familiarity in some conditions. (and I'm not confident whether the perceived familiarity is a part of search interactions.)

Overall, I think that this is a solid work that investigate the effects of confirmation bias in multi-query sessions on users’ attitudes and search interactions. However, I have a little hesitation to fully support the acceptance of this work due to the two experimental setting problems.

---

### Official Review · Reviewer_nhc8 · 2024-05-21

**Rating:** 0
**Confidence:** 4

**Objective Part Of Review:**

The paper presented a user study on how users interact with search results in three query sessions on debate topics.   Here users are assumed to have cognitive bias while the results are algorithmic biased.

The problem is clearly stated, and the methodology is well designed.   Here are some concerns.
* The data set consists only 4 debate topics, which are on less controversial topics.
* Users need to select queries from an existing pool, can are unable to formulate their own queries.

The research questions are clearly listed, and answers are discussed.

The paper mentioned "they first selected a query term from a per-defined query set", but Figure 1 seems suggested the queries can have more than one term.

**Subjective Part Of Review:**

The paper is easy to follow.   The experiment designs are clearly described.

One main concern is how users formulate the queries.  It is unclear why the users have to pick the ones from the existing lists, rather than formulating the results by themselves.

Another concern is the scope of the study is rather limited,  focusing on only 4 non-controversial topics.   It would be more interesting to see the conclusions on more controversial topics and check the consistencies.


The submissions would fit CHIIR better than ICTIR.

---

### Meta-Review · Area_Chair_wCwv · 2024-05-30

**Recommendation:** Reject
**Confidence:** 4

**Metareview:**

The submission presents a study of how cognitively biased users interact with biased search results, elucidating the effects of confirmation bias and bias mitigation. The study has been conducted based on four mildly controversial topics and users have to engage in a three-query search session in which they select predefined queries. Participants of the study were recruited on MTurk and screened for having biased opinions at least on one of the four topics.

All four reviewers pointed out several weaknesses with the submission. First and foremost, given that this is a purely empirical work, it appears that ICTIR is not the right venue. Second, some of the design choices of the study should be revised or better motivated (e.g., the consideration of only four topics and that users had to select queries from a predefined pool). Third and last, there are some minor yet fixable issues with the presentation.